# Failure Prediction Is a Better Performance Proxy for Early-Exit Networks Than Calibration

## Abstract

Early-exit models speed up inference by attaching internal classifiers to intermediate layers of the model and allowing computation to stop once a prediction satisfies an exit criterion. Most early-exit methods rely on confidence-based exit strategies, which motivated some works to calibrate intermediate classifiers to improve the performance of the entire model. In this paper, we show that calibration measures can be misleading indicators of the performance of multi-exit models: a well-calibrated classifier may still waste computation, and common calibration methods do not preserve the sample ranking within a classifier. We demonstrate empirical cases where miscalibrated networks outperform calibrated ones. As an alternative, we propose to use failure prediction as a more useful proxy for early-exit model performance. Unlike calibration, failure prediction accounts for changes in the ranking of samples and shows a strong correlation with efficiency improvements, making it a more dependable basis for designing and evaluating early-exit models.

## 1 Introduction

The rapid growth of deep learning increases the demand for resource-efficient models. Early-exit models address this challenge by attaching classifiers to intermediate model layers and enabling the model to save computation by stopping the inference once a prediction satisfies an exit criterion. Early-exits were initially introduced for vision models [6, 9, 20, 26], and have since then become a natural fit for resource-constrained scenarios [2, 3, 12, 13, 24, 25, 28]. More recently, they also have been successfully adopted for natural language processing [7, 14, 23, 29, 35], including reasoning models [8, 31] where inference efficiency is critical.

The most common approach to exiting uses prediction confidence and enables the network to halt early if an intermediate classifier produces a sufficiently confident prediction. Consequently, many works focused on improving the calibration of the classifiers [15, 17–19, 21], under the assumption that better calibration yields better models. We challenge that assumption and argue that calibration can be a misleading proxy for early-exit models. We demonstrate cases where miscalibrated networks behave counterintuitively and outperform calibrated multi-exit models, as demonstrated in Figure 1. Finally, we highlight that calibration methods can introduce unintended side effects, such as altering the maximum confidence ranking of samples within the classifier.

Given the above-mentioned issues, we propose to use *failure prediction* [1] as a more suitable proxy for early-exit network performance. We discuss its desirable properties, showing that, unlike calibration, failure prediction measures are sensitive to changes in rankings of samples. Finally, we adapt failure prediction measures to the multi-exit setting by defining the *Early-Exit Failure Prediction score* (EEFP score). Notably, our experiments demonstrate that our proposed EEFP score shows a strong correlation with early-exit performance in cases where calibration measures do not.

Submitted to 39th Conference on Neural Information Processing Systems (NeurIPS 2025). Do not distribute.

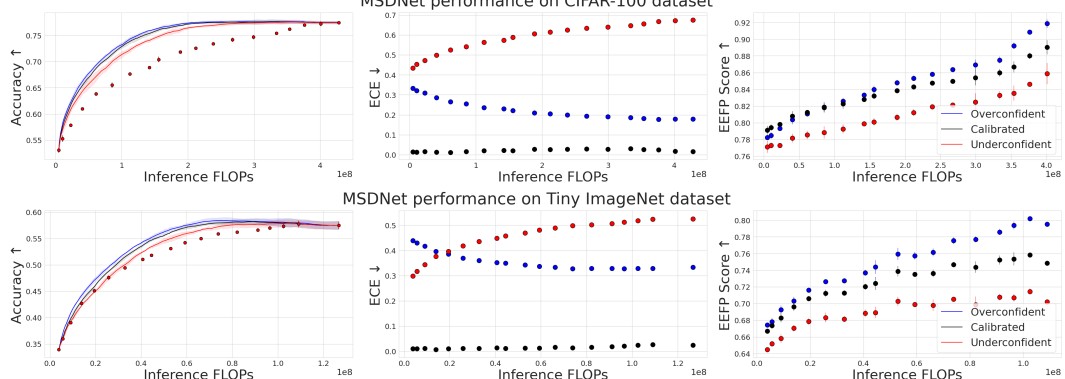

Figure 1: Cost-accuracy curves, head calibration errors, and our proposed EEFP scores for one calibrated model and two decalibrated models with modified temperature values. Calibration fails to capture the quality of the early-exit model, as an overconfident network with higher ECEs performs better than the calibrated one. We propose an alternative metric, *Early-Exit Failure Prediction score* (EEFP score), which more accurately reflects the quality of the multi-exit model.

We hope that our work advances the discussion on appropriate evaluation metrics for early-exit models and helps improve the design of future methods for efficient computation.

## 2   Background

**Early-exits**   We consider the standard early-exit framework used in prior work [6, 9], and start with a multi-exit model architecture with $J$ classifiers. Each classifier $g_j$, $j \in \{1, \ldots, J\}$ maps an input $x$ to a probability vector over $C$ classes: $p_j = g_j(x)$. The network evaluates classifiers sequentially until the *confidence* of the prediction (usually the maximum probability obtained via softmax) exceeds the corresponding exit threshold, that is: $c_j(x) = \max_k p_{j,k} \geq \tau_j$, where $k = 1, \ldots, C$ refers to class indices [7]. If the exit condition is satisfied, the network outputs $p_j$; otherwise, evaluation continues until another classifier exits or the final classifier $g_J$ is reached. By varying the exit thresholds, we obtain a *compute–accuracy trade-off curve*, where compute is measured as the average of floating-point operations per sample [30, 32].

**Calibration**   Confidence calibration refers to how well a model's predicted confidence matches its actual accuracy [4]. In a well-calibrated model, predictions made with high confidence are more likely to be correct. Calibration is a key property of probabilistic models and has been widely studied in the context of model reliability and trustworthiness [4, 10, 22, 27]. The calibration of a given classifier can be measured via expected calibration error (ECE). To calculate ECE, one partitions predictions into $M$ bins $B_1, \ldots, B_M$ and calculates the accuracy $\mathrm{acc}(B_m)$ and average confidence $\mathrm{conf}(B_m)$ in each bin $m$. Then, ECE can be calculated as:

$$\mathrm{ECE} = \sum_{m=1}^{M} \frac{|B_m|}{n} \left| \mathrm{acc}(B_m) - \mathrm{conf}(B_m) \right|.$$

## 3   Calibration in early-exit models

In context of early-exiting, prior work [17, 19] linked the calibration of the intermediate classifiers with improved performance of the model. Since calibration changes head's confidence, it can alter sample distribution across exit points and thus impact the overall model performance. Meronen et al. [17] explicitly focus on improving early-exit performance via calibration, stating that "adequately quantifying and accounting for (...) uncertainty improves the predictive performance and aids decision-making". Similarly to Pacheco et al. [19] and Wójcik et al. [29], they measure the calibration error of each classification head. Despite the existing body of work mentioned above, **we hypothesize**

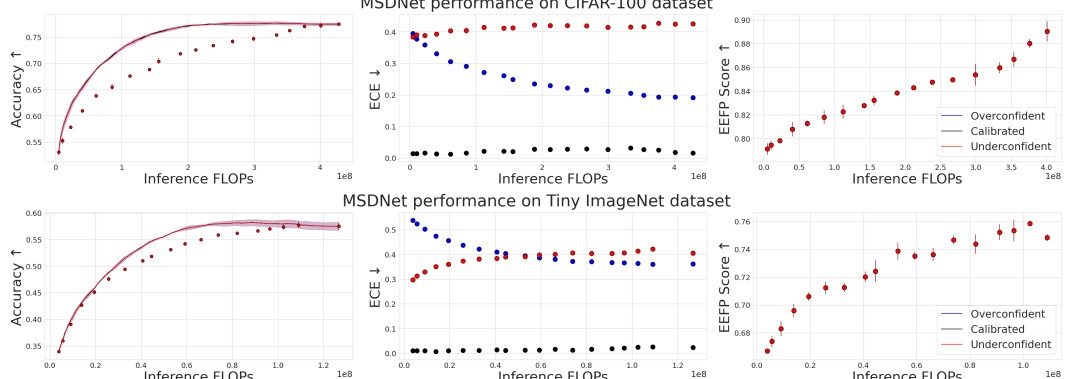

Figure 2: Performance of models decalibrated using a transformation that preserves the ranking of samples within each classifier. Despite significantly deteriorated ECE scores, model performance remains unchanged, with all three models showing almost identical cost–accuracy curves. EESP score perfectly reflects this behavior, assigning identical score values to all three networks.

**that calibration does not correlate with the performance of the early-exit networks, and may even hurt it in practice**. In the following sections, we provide empirical evidence supporting this hypothesis by showing that calibration of early-exit networks can affect performance in unexpected ways. To this end, we adopt the setting from Meronen et al. [17] with the MSDNet multi-exit architecture [6], and find the per-IC thresholds for every considered budget after the training and (de)calibration via the procedure proposed by Huang et al. [6]. We use cost-accuracy curves to compare the network performance, and mark the performance of individual classifiers with dots. Please refer to Appendix E for the experimental details, and to Appendix C for the additional experimental results with alternative calibration methods.

### 3.1 Better calibration does not always lead to better performance

In our first experiment, we calibrate the intermediate classifiers of a trained MSDNet model with temperature scaling [4]. Subsequently, we create two decalibrated variants of the same model by multiplying the temperature used in each head by 3.0 or 0.3, which results in an underconfident or overconfident model, respectively. We evaluate both accuracy and expected calibration error (ECE) of the three models across varying computational budgets, and present the results in Figure 1. Our experiments reveal that **models can achieve a more favorable cost–accuracy trade-off, despite exhibiting substantially worse calibration scores.** This somewhat counterintuitive finding indicates that **calibrating intermediate classifiers might not be beneficial for multi-exit models, which is contrary to what was suggested in previous studies** [17–19, 29].

### 3.2 Worse calibration does not always lead to worse performance

To further analyze the relationship between head calibration and the final performance of the model, we devise the following experiment. Instead of manipulating the temperature of each head, we propose an even simpler decalibration method, which directly transforms the final confidence estimate of the classifier: $\hat{c}_j(x) = \frac{1}{C} + \left(1 - \frac{1}{C}\right) \cdot \left(\frac{c_j(x) - \frac{1}{C}}{1 - \frac{1}{C}}\right)^{\alpha}$ , where $\alpha$ is the *decalibration coefficient* and $C$ refers to the number of classes. Formally, for any classifier $j$ let $\pi_j$ be a permutation of sample indices such that: $c_j(x_{\pi(1)}) > c_j(x_{\pi(2)}) > \cdots > c_j(x_{\pi(n)})$ (see Appendix B for a more detailed discussion on the design of our function). The $\hat{c}_j$ **transformation preserves the order of confidences between the samples (*sample ranking*). The same is not true for temperature scaling** [1], a fact that we describe in detail in Appendix A.

We perform the alternative decalibration experiment by using $\hat{c}_j$ to decalibrate the classifiers. We set $\alpha$ to 10 and 0.1, obtaining an underconfident and overconfident model, respectively. The results

---

[1]Consider two sets of logits: $l^1 = [0.65, 0.34, -1.03]$, $l^2 = [-0.06, -0.11, 0.60]$. If we compute confidence $c$ with the softmax function, for $T_1 = 1.0$: $c^1 > c^2$, a while for $T_2 = 0.3$: $c^1 < c^2$.

are shown in Figure 2. **This time, despite decalibrating the model significantly, the overall accuracy-cost performance remains the same, even for the underconfident model.** This further illustrates that calibration metrics do not necessarily reflect actual model performance, and do not capture the important aspect of sample rankings.

## 4    Early-exit failure prediction

The ranking-preserving aspect of the calibration methods proved to be essential in our previous experiments. We observe that the **separability between correctly and incorrectly classified samples** may be a more reliable indicator of early-exit model performance than calibration measures. In the literature, this separability is directly related to *failure prediction* (or *error and success prediction*) [1, 5]. Crucially, prior work has shown that calibrating classifiers can actually degrade failure prediction performance [36].

To evaluate failure prediction for a single classifier, we record the classifier's confidence score for each sample and label it as $y = 1$ if the prediction is correct and $y = 0$ otherwise. We then compute the area under the receiver operating characteristic curve (AUROC), which measures the probability that a randomly chosen correct prediction ($y = 1$) receives a higher confidence score than a randomly chosen incorrect one ($y = 0$), thus reflecting how well the classifier's confidence separates correct from incorrect predictions.

However, this definition of failure prediction was devised for conventional static classifiers, and is not suitable for early-exit networks, as it does not account for the behavior of deeper classifiers. In particular, if a sample is incorrectly classified by the current classifier and all of the deeper classifiers, then it is beneficial to halt computation as early as possible. This crucial observation leads us to adapt the definition of failure prediction to the multi-exit model setup. **We define Early Exit Failure Prediction score (EEFP score)** as:

$$\text{EEFP}_j(\{x_i, y_{j,i}\}) = \text{AUROC}(\{c_j(x_i), \bar{y}_{j,i}\}),$$

where:

$$\bar{y}_{j,i} = \begin{cases} 1, & \text{if } y_{j,i} = 1 \lor (y_{j,i} = 0 \land \forall_{l>j} y_{l,i} = 0) \\ 0, & \text{otherwise.} \end{cases}$$

In this formulation, a positive $y_{j,i}$ means that either the current head is correct, or all deeper heads would also be wrong, and exiting is optimal from the computational point of view.

We report the EEFP scores for each classifier head alongside the results of our previous experiments from Sections 3.1 and 3.2 in Figures 1 and 2. In both cases, the EEFP scores correlate well with the overall accuracy-cost performance of the model. When performing temperature decalibration, the EESP score is the highest for the overconfident model, which actually performs better but achieves worse ECE. In the case of rank-preserving decalibration, EESP scores remain constant and reflect the identical performance of all three investigated models. **EEFP score better reflects both prediction accuracy and the effect of early exits on the model performance, making it more suitable for evaluating early-exit models than standard calibration metrics**.

## 5    Conclusion

Our work challenges the common assumption about positive effects of calibration of intermediate classifiers in the early-exit models. Through a series of controlled experiments, we demonstrated that calibration metrics such as ECE can be misleading: well-calibrated models may still waste computation, while deliberately miscalibrated models can sometimes achieve better cost–accuracy trade-offs. A key factor behind this discrepancy is that calibration metrics fail to capture the separability of correctly and incorrectly classified samples, which directly influences the efficiency of the early exits.

To address these limitations, we propose the Early-Exit Failure Prediction Score (EEFP Score), a failure prediction metric specifically tailored to the multi-exit setting. Unlike calibration metrics, EEFP Score directly measures how well a model distinguishes between samples that benefit from further computation and those that do not. EEFP Score better reflects the real goals of efficient inference, and we found that it strongly correlates with early-exit performance across the experimental settings where calibration metrics fail to accurately capture model quality.

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

# Appendix

## A Influence of temperature scaling

### A.1 Probability distribution after temperature scaling

In a classification problem with $C$ classes, suppose the model outputs logits $z_1, \ldots, z_C$ for a given data sample, and let $r$ denote the index of the most probable class. Without temperature scaling, the softmax probabilities are

$$p_k = \frac{e^{z_k}}{\sum_{l=1}^{C} e^{z_l}}, \quad k = 1, \ldots, C.$$

By introducing

$$d := \log\left(\sum_{l=1}^{C} e^{z_l}\right),$$

we can equivalently write

$$p_k = e^{z_k - d}, \quad \text{and hence} \quad z_k = \log(p_k) + d.$$

When scaling logits by a temperature parameter $T > 0$, the probabilities become

$$p_k^{(T)} = \frac{e^{z_k/T}}{\sum_{l=1}^{C} e^{z_l/T}} = \frac{p_k^{1/T}}{\sum_{l=1}^{C} p_l^{1/T}}, \quad k = 1, \ldots, C.$$

Therefore, the confidence changes from $p_r$ to

$$p_r^{(T)} = \frac{p_r^{1/T}}{\sum_{l=1}^{C} p_l^{1/T}}.$$

Since the denominator $\sum_{l=1}^{C} p_l^{1/T}$ depends on the entire probability distribution (and not solely on $p_r$), two samples with the same original confidence $p_r$ can yield different scaled confidences $p_r^{(T)}$ after temperature scaling.

### A.2 Temperature scaling does not preserve the ranking of samples

Table 1: Classifier's logits in a toy problem.

|          | Class 1 | Class 2 | Class 3 | Class 4 |
|----------|---------|---------|---------|---------|
| Image $A$ | -0.7985 | -0.9163 | -2.3026 | -2.9957 |
| Image $B$ | -1.6094 | -1.6094 | -0.9163 | -1.6094 |
| Image $C$ | -1.2040 | -0.9676 | -1.3471 | -2.8134 |

Table 2: Classes probability distribution (after softmax) in a toy problem.

|          | Class 1 | Class 2 | Class 3 | Class 4 |
|----------|---------|---------|---------|---------|
| Image $A$ | 0.450 | 0.400 | 0.100 | 0.050 |
| Image $B$ | 0.200 | 0.200 | 0.400 | 0.200 |
| Image $C$ | 0.300 | 0.380 | 0.260 | 0.060 |

Consider a toy example with four classes. The $j-th$ classifier's logit outputs are shown in Table 1, while the corresponding softmax probabilities are reported in Table 2. Confidences of the predictions are as follows: $c_j(A) \approx 0.450$, $c_j(B) \approx 0.400$, $c_j(C) \approx 0.380$. Therefore the ranking of samples is $A, B, C$ (from the most to the least confident ones). However, when temperature changes, the ranking does as well. For example for temperature 0.3, $c_j(A) \approx 0.594$, $c_j(B) \approx 0.771$, $c_j(C) \approx 0.575$, and the ranking changes to $B, A, C$. On the other hand, for temperature 3.0, $c_j(A) \approx 0.328$, $c_j(B) \approx 0.296$, $c_j(C) \approx 0.299$, and the ranking changes to $A, C, B$.

## B  Monotonic decalibration function

We define rank-preserving decalibration transformation from Section 3.2 as:

$$f_\alpha(c) \;=\; \frac{1}{C} \;+\; \left(1 - \frac{1}{C}\right) \left(\frac{c - \frac{1}{C}}{1 - \frac{1}{C}}\right)^\alpha,$$

and use it to obtain $\hat{c}_j(x) = f_\alpha(c_j(x))$. In $f_\alpha$, $C$ is constant, and $c \;\mapsto\; \frac{c - \frac{1}{C}}{1 - \frac{1}{C}}$ increases with increasing $c$. Raising a positive increasing function to the power $\alpha > 0$ also preserves monotonicity. Therefore, $f_\alpha(c)$ is strictly increasing with increasing $c$.

However, due to numerical reasons, $f_\alpha$ may increase so slowly that finite precision arithmetic makes it appear non-monotonic. To avoid this, we introduce a slightly modified function:

$$\hat{f}_\alpha(c) \;=\; \epsilon c \;+\; (1 - \epsilon)\, f_\alpha(c),$$

where $\epsilon = 5 \cdot 10^{-2}$. This guarantees that, on any sufficiently long interval $[c_1, c_2]$, the increase of $\hat{f}_\alpha$ is large enough to remain distinguishable under finite numerical precision.

Moreover, $\hat{f}_\alpha(\frac{1}{C}) = \frac{1}{C}$ and $\hat{f}_\alpha(1) = 1$, which corresponds to the cases where all classes are equally probable or where one class has probability one, respectively. Therefore, we ensure that the confidence values remain within the range that would also be possible under the standard definition.

## C   Temperature decalibration for models calibrated with alternative methods

To ensure our findings are not specific to a single training or calibration setup, we analyze the effect of temperature decalibration on model accuracy in the setting proposed by [17]. They consider two approaches that improve model calibration: last-layer Laplace approximations (LAP) [11] and model-internal ensembles (MIE). We test these approaches using three MSDNet variants with 4, 6, and 8 blocks (referred to as "Small," "Medium," and "Large" in our plots) trained on CIFAR100, and analyze the impact of temperature decalibration on the models in the following subsections.

### C.1   LAP

We begin with a model calibrated using LAP, where we perform a grid search over the temperature and Laplace prior variance to obtain the best-calibrated model. We then decalibrate the model by applying temperature modifications to its predictions, as in the main paper. We show the corresponding results in Figure 3. Likewise, the performance of the declibrated model does not degrade and is even slightly better than the performance of the calibrated one.

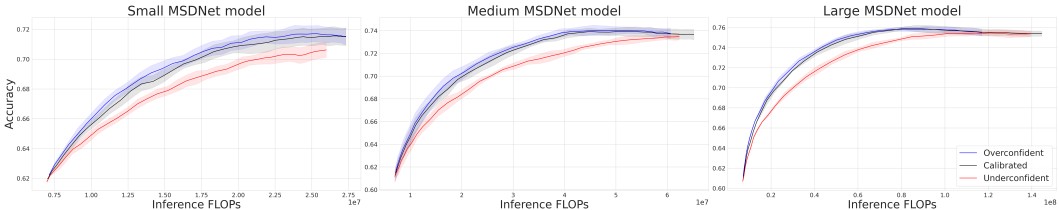

Figure 3: Results for LAP-calibrated MSDNet models.

### C.2   LAP+MIE

Secondly, we combine LAP with MIE following Meronen et al. [17], and apply post-hoc temperature decalibration as before. We show the results in Figure 4. The results are consistent with our main findings and all other experiments: despite being decalibrated, the overconfident model does not suffer and even slightly outperforms the calibrated model. Together with the results from the previous section, these findings demonstrate that our conclusions hold beyond a specific training or calibration setup and are consistent across both calibration methods and multiple model sizes.

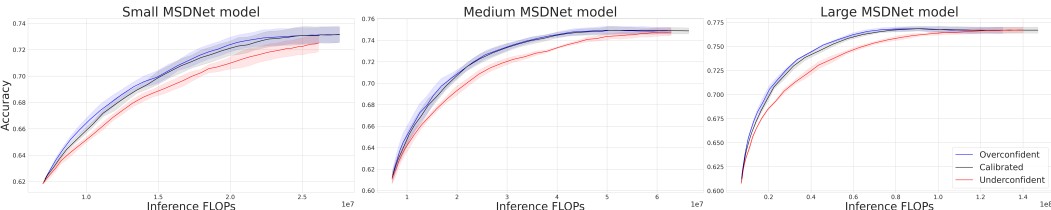

Figure 4: Results for MIE-calibrated MSDNet models.

# D Alternative measure for early-exit failure prediction

EEFP Score measures the separability of two subsets (correctly and incorrectly predicted) for every possible threshold. However, it is unreasonable to assume that every threshold is equally important for early-exit models, and in practice, we care about a subset of thresholds. For example, for high computational budgets, the first threshold $\tau_1$ tends to be close to 1.0, and separability on low thresholds $\tau_1 \to 0$ is irrelevant. Moreover, EEFP Score is threshold-agnostic and is measured for each exit separately.

In this section we consider an alternative way to estimate failure prediction, the **EEF1** score, which uses F1 instead of AUROC. EEF1 can be used to compare two multi-exit models for any budget by using the actual exit criterion threshold used during inference.

Let $\tau_j$ denote the threshold of the $j$-th classifier. We define an exit indicator as:

$$h_j(x) = \begin{cases} 1, & \text{if } c_j(x) \geq \tau_j, \\ 0, & \text{otherwise.} \end{cases}$$

Let $\mathcal{L}_j$ be the set of indices for which the model has not exited before the $j$-th classifier:

$$i \in \mathcal{L}_j \iff \forall_{l \in \{1,\ldots,j-1\}} \, c_l(x_i) < \tau_l.$$

We define the **Early Exit F1 Score** for the $j$-th classifier as:

$$\text{EEF1}_j(\{x_i, y_{j,i}\}_{i \in \mathcal{L}_j}) = \text{F1}(\{h_j(x_i), \bar{y}_{j,i}\}_{i \in \mathcal{L}_j}).$$

To obtain a single score for each budget, we compute the arithmetic mean over all classifiers:

$$\text{EEF1}(\{x_i, y_{j,i}\}) = \frac{1}{J} \sum_{j=1}^{J} \text{EEF1}_j(\{x_i, y_{j,i}\}_{i \in \mathcal{L}_j}).$$

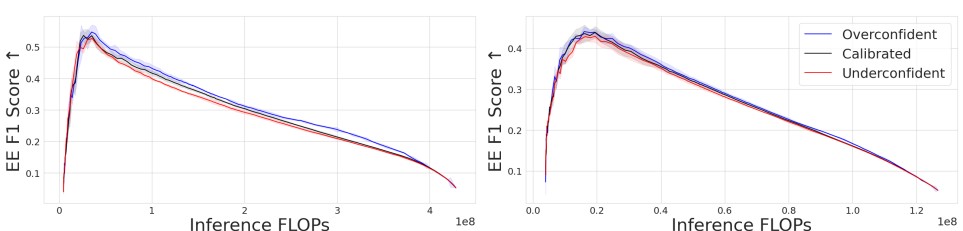

Figure 5: EEF1 scores for CIFAR-100 (left) and Tiny ImageNet (right)

The resulting EEF1 scores for models decalibrated via modified temperature (see Section 3.1) are shown in Figure 5. Like the EEFP scores, EEF1 indicates a small advantage for the overconfident model compared to the calibrated model. However, EEFP better separates the models' failure-prediction performance and is more straightforward to interpret; for these reasons, we decided to focus on EEFP score in the main paper.

## E  Experimental setup details

In this section, we describe the experimental setup used in our experiments in the main paper.

### E.1  Architecture

We adopt the MSDNet architecture [6] with 7 blocks. We attach early-exit heads after each block, and additionally inside the blocks.

### E.2  Training

For each experiment, we train three models with independent initializations and report the average performance across these three seeds. We train MSDNet models with a batch size of 512, using a learning rate of 1e-3 and AdamW optimizer [16] without weight decay. We use a cosine annealing scheduler with warm restarts and linear warm-up. As data augmentations, we apply random resizing, cropping, rotation, contrast adjustment, random erasing, Mixup [34], and CutMix [33]. All models are trained until convergence.

### E.3  Calibration

To calibrate the early-exit model, we proceed with a gradient-based approach. We extract 2.5% of the samples from the training set, which were not used to optimize the model during training, and use them for the calibration phase. We then freeze the model's parameters, attach temperature scaling calibrators, and proceed by minimizing NLPD. Each early-exit head is optimized individually and has its own temperature.

### E.4  Evaluation

During evaluation, we begin with the calibrated model and additionally consider two variants: an overconfident and an underconfident model. All three models share the same trained parameters and differ only in how the logits are transformed.

For a given data sample $x$, let

$$z_{j,1}, \ldots, z_{j,C}$$

denote the logits of the classifier in the calibrated model. In the temperature scaling experiment 3.1, the logits of the models are modified by a temperature parameter $T$ as follows:

$$\frac{z_{j,1}}{T}, \ldots, \frac{z_{j,C}}{T}.$$

Softmax is then applied to each set of logits, and the confidence values are obtained from the maximum softmax probability, denoted as $c_j(x)$. In Section 3.2, instead of scaling the logits, the confidence is derived using the function $\hat{c}_j(x)$ in place of the standard $c_j(x)$.

For each model and each exit head, decision thresholds are determined using the validation set. To this end, we follow the threshold selection heuristic proposed in Huang et al. [6], Meronen et al. [17], which derives appropriate thresholds based on the distribution of confidence scores.

Given a predefined FLOPs budget, the network is expected to terminate at each exit for a certain fraction of the input samples. This allocation of samples across exits is controlled by a parameter $q$. At the $j$-th exit, the model is required to terminate for a fraction of samples defined as:

$$\text{exit-share}(j) = \frac{q^j}{\sum_{l=0}^{J-1} q^l} \tag{1}$$

During inference on the test set, each model computes its own confidence values and applies its own set of thresholds to decide when to exit and, therefore, which prediction to make.

