# OpenReview forum: "Failure Prediction Is a Better Performance Proxy for Early-Exit Networks Than Calibration"
_NeurIPS.cc/2025/Workshop/Reliable_ML — NeurIPS 2025 - Reliable ML Workshop_

### Official Review · Reviewer_JWDu · 2025-09-07
**Novel framework of failure prediction as a proxy for early-exit networks**

**Rating:** 8
**Confidence:** 4

**Review:**

This paper proposes replacing confidence calibration with failure prediction as the main performance proxy for early-exit networks. The idea is insightful and well-motivated, as it challenges prevailing assumptions and reframes the problem from calibration toward separability of correct vs. incorrect samples. The paper’s contribution is timely, and the empirical evidence convincingly supports the claim that calibration does not always correlate with early-exit efficiency.

**Strengths**
1. Changing the perspective from calibration to failure prediction is original and intriguing.
2. The empirical evidence (including controlled experiments with calibrated vs. decalibrated models) strongly supports the central hypothesis.
3. The introduction of the EEFP score is novel and aligns well with the practical goals of early-exit networks.

**Weaknesses**
1. The hypothesis that calibration may even hurt early-exit performance is supported empirically, but the paper does not explain why this phenomenon occurs. A deeper analysis of underlying causes would strengthen the contribution.
2. Details of the baselines are underdeveloped. More description of dataset choices, training settings, and competing methods is needed for reproducibility and fair comparison.
3. Figure 2, which represents the main experimental evidence, is difficult to interpret. The legends and presentation should be reorganized for clarity.
4. It remains unclear how the EEFP score can be used at test time: how are thresholds determined, and how would practitioners apply the metric in practice?

---

### Official Review · Reviewer_95hN · 2025-09-19
**Interesting alternative approach for early-exit networks**

**Rating:** 5
**Confidence:** 3

**Review:**

The paper tackles the problem of how to design a suitable criterion for early-exit networks. As the authors mention, this was achieved by prediction confidence (would be useful to highlight citations here), and calibration. In this work, the authors revisit an older approach termed failure prediction, and define a score termed the EEFP which is based on the failure prediction.


Some questions that I have, which I feel could be have been addressed in the paper:
1. What are the markers in the plots in Figure 1? How does Figure 1 help understand the pitfalls of calibration for early-exit?
2. What is the purpose of having J classifiers? Are all of them trained simultaneously, and are the J classifiers meant to be viewed as J different heads of a base neural networks, or are these "spawned" from intermediate layers?
3. Why is the ECE a good metric? How are the bins chosen?
4. Should the flops curves be a step function roughly (since there is a constant number of operations in between two classifier heads)?

I feel like this is an interesting paper, but the problem could be better elucidated; as a reader who is not familiar with this literature, there were parts of the paper that was difficult to follow.

---

### Official Review · Reviewer_6NhV · 2025-09-19
**Marginally above acceptance threshold**

**Rating:** 6
**Confidence:** 2

**Review:**

## Summary

This work compares the results of Failure Prediction and Calibration for  Early-Exit Networks, which aims to speed up the inference. This work proposes situations in which calibration cannot work as a successful proxy and suggests using early exit failure prediction as an alternative.

## Strengths

1. This work shows the failure cases in which the Calibration fails for the proxy.
2. The paper is well-written and easy to follow.
3. This paper focuses on an important question.

## Weakness

1. The failure cases for the Calibration are artificially created, and we all know no method can be applied to all methods.
2. Without larger-scale experiments, it's difficult to claim the EEFP score is a better proxy than the Calibration. To prove this on a larger scale and conduct experiments with diverse datasets is necessary.

## Suggestions for Authors
1. See the weakness.